# Evaluating Prescription Pattern and Effectiveness of Antihypertensive Drugs in Non-Operated Aortic Dissection Patients

**DOI:** 10.3390/jcm12051962

**Published:** 2023-03-01

**Authors:** Yun-Hui Huang, Kai-Lin Chiu, Chuan-Wei Shen, Ming-Jong Bair, Chung-Yu Chen

**Affiliations:** 1School of Pharmacy, Kaohsiung Medical University, Kaohsiung 80716, Taiwan; 2Department of Pharmacy, Taipei Tzu Chi Hospital, Buddhist Tzu Chi Medical Foundation, New Taipei City 23142, Taiwan; 3Department of Pharmacy, Kaohsiung Medical University Hospital, Kaohsiung 80716, Taiwan; 4Mackay Medical College, New Taipei City 25245, Taiwan; 5Division of Gastroenterology, Department of Internal Medicine, Taitung Mackay Memorial Hospital, Taitung 95054, Taiwan; 6Department of Medical Research, Kaohsiung Medical University Hospital, Kaohsiung 80716, Taiwan

**Keywords:** aortic dissection, effectiveness, antihypertensive drugs

## Abstract

Introduction: Aortic dissection (AD) is a life-threatening disease. However, the effectiveness of different strategies of antihypertensive therapies in non-operated AD patients is still unclear. Materials and methods: Patients were classified into five groups (groups 0–4) based on the number of classes of antihypertensive drugs, including β-blockers, renin-angiotensin system (RAS) agents (angiotensin-converting enzyme inhibitors (ACEIs), angiotensin II receptor blockers (ARBs), and the renin-inhibitors), calcium channel blockers (CCBs), and other antihypertensive drugs, were prescribed within 90 days after discharge. The primary endpoint was a composite outcome of re-hospitalization associated with AD, referral for aortic surgery, and all-cause death. Results: A total of 3932 non-operated AD patients were included in our study. The most prescribed antihypertensive drugs were CCBs, followed by β-blockers and ARBs. Within group 1, compared to other antihypertensive drugs, patients using RAS agents (aHR, 0.58; *p* = 0.005) had a significantly lower risk of occurrence of the outcome. Within group 2, the risk of composite outcomes was lower in patients using β-blockers + CCBs (aHR, 0.60; *p* = 0.004) or CCBs + RAS agents (aHR, 0.60; *p* = 0.006) than in those using RAS agents + others. Conclusion: For non-operated AD patients, RAS agents, β-blockers, or CCBs should be given in a different strategy of combinations to reduce the hazard of AD-related complications compared to other agents.

## 1. Introduction

Aortic dissection (AD) is a life-threatening disease with a low incidence rate but a high mortality rate [1]. Patients with AD are assessed to receive surgical treatment based on the location of the dissection, aortic diameter, expansion rate, or AD complications. The outcome of AD is dependent on the different types of AD and the management the patients received [2,3]. A national study reported that in non-operated Type B AD patients, the associated mortality rate was lower during hospitalization but became higher after discharge compared to patients who underwent surgery [4].

Medical therapy plays a crucial role in slowing down the progression of expansion or rupture of the aortic wall. β-blockers are recommended as first-line drugs for AD treatment in the acute and chronic phases of the condition per the guidelines from the American College of Cardiology Foundation (ACCF)/American Heart Association, the European Society of Cardiology, and the Japanese Circulation Society [5,6,7,8]. However, there is a paucity of supporting studies; thus, the level of evidence is still limited. In addition, patients are often prescribed multiple antihypertensive agents to achieve optimal control in clinical practice. Nevertheless, most published studies evaluate the effectiveness of certain antihypertensive medications without considering the combined use of other antihypertensive drugs. Therefore, this nationwide retrospective cohort study aimed to investigate the prescription pattern and effectiveness of different combinations of antihypertensive drugs in non-operated or non endovascular aortic dissection repair AD patients, including type B AD or type A AD.

## 2. Materials and Methods

### 2.1. Data Source

We conducted a population-based retrospective cohort study using data from the Taiwan National Health Insurance Research Database (NHIRD) from January 2011 to December 2019. The National Health Insurance program which was launched in 1995 covers about 99.6% of the Taiwanese population, and its database provided information on disease diagnosis, medications, medical procedures, and admission records containing outpatient, inpatient, and emergency visits [9]. The study was approved by the institutional review board of the Kaohsiung Medical University Chung-Ho Memorial Hospital (KMUHIRB-E(I)-20210272), which waived the informed consent due to the retrospective nature of the study. Patients’ privacy was ensured by encrypting information on patient identification, medical institutions, and health professionals. The study was independently conducted at a subcenter of the Health and Welfare Data Science Centers at Kaohsiung Medical University.

### 2.2. Study Population

Newly diagnosed non-operated or non endovascular aortic dissection repair AD patients aged > 20 years were identified via inpatient visit records from 1 January 2012 to 31 December 2017 (*International Classification of Diseases, Ninth Revision* (ICD-9-CM) codes 441.00–441.03 and *Tenth Revision* (ICD-10-CM) codes I71.00–I71.03). AD patients were symptomatic and received inpatient treatment during their initial hospitalization; however, the study participants did not undergo aortic dissection surgery or endovascular aortic repair at any location. Non-operated AD patients included type B AD or type A. After discharge, they were recommended to have at least one outpatient visit within one month and then every three months thereafter. Therefore, we excluded patients who died during hospitalization and set a 90-day period after discharge to monitor the prescription of antihypertensive drugs during outpatient visits.

Patients with aortic aneurysms or congenital connective tissue disorders such as Marfan syndrome were excluded from the study. Patients who did not undergo computed tomography, transesophageal echocardiography (TEE), or magnetic resonance imaging were also excluded.

### 2.3. Drug Use and Prescription Pattern

Patients were classified into five groups (groups 0–4) based on the number of classes of antihypertensive drugs that were prescribed during the 90-day post-discharge period as follows: (1) group 0 was composed of patients with no exposure to antihypertensive drugs, (2) group 1 was composed of patients who were prescribed antihypertensive drugs of the same class, (3) group 2 was composed of patients who were prescribed antihypertensive drugs of two different classes, (4) group 3 was composed of patients who were prescribed antihypertensive drugs of three different classes, (5) group 4 was composed of patients who were prescribed antihypertensive drugs of four different classes. Antihypertensive drugs were classified into four main classes in this study, including β-blockers, calcium channel blockers (CCB), renin-angiotensin system (RAS) agents (angiotensin-converting enzyme inhibitors (ACEIs), angiotensin II receptor blockers (ARBs), and renin-inhibitors), and other antihypertensive drugs (diuretics, hydralazine, minoxidil, nitroprusside, centrally α2-agonists, α-blockers, reserpine, Rauwolfia serpentine, guanethidine). Patients without any prescription of antihypertensive drugs or with less than 28 days of accumulated antihypertensive drug prescription were defined as non-drug users (group 0). Besides the number of classes, we also monitored the combination of antihypertensive drug prescription patterns during the 90-day post-discharge period in these patients.

### 2.4. Outcomes

In this study, the primary endpoint was a composite outcome of re-hospitalization associated with AD, referral for aortic surgery, and all-cause death, stratified by groups 1–4. Upon multiple comparisons between the different groups, we found that the reference control group was different. Group 1 was composed of patients who took other antihypertensive drugs, group 2 was composed of those who took RAS agents combined with other agents, and group 3 was composed of patients who took a triple combination (CCB, RAS, and others) as a reference control group. Patients were followed up from the index discharge date until the occurrence of composite outcome or the last date of the database (31 December 2019), whichever came first. Patients who did not experience adverse events during the follow-up period were censored.

### 2.5. Characteristics

Patients’ clinical characteristics included age, sex, geographic area, urbanization, insurance premium, comorbidities, Charlson comorbidity index (CCI), and inpatient comedications including antiplatelets, anticoagulants, antidiabetic agents, and statins. The CCI score is a method of categorizing comorbidities of patients based on ICD. Each comorbidity category has an associated weight (from 1 to 6) and a summed score of 19 comorbidities weighted according to severity.

The locations of aortic dissections were classified according to ICD codes, stratified by unspecified aortic dissection sites (UAD), thoracic aortic dissection (TAD), abdominal aortic dissection (AAD), and thoracoabdominal aortic dissection (TAAD). We identified comorbidities with at least two outpatient diagnoses or one inpatient diagnosis at the index admission and the previous year, according to the code of ICD-9-CM or ICD-10-CM.

### 2.6. Statistical Analysis

Continuous variables were presented as means (standard deviations) and categorical variables were presented as frequencies (percentages). Univariate and multivariable Cox proportional hazards models were used to estimate the association between the primary outcome and different combinations of antihypertensive drug prescription patterns, presented with hazard ratios and 95% confidence intervals (CIs). The multivariable Cox proportional hazards model adjusted covariates selected by stepwise multiple regression analyses and important risk factors associated with AD, including age, sex, comorbidities of hypertension, hyperlipidemia, diabetes mellitus, heart failure, coronary artery disease, cerebrovascular disease, chronic kidney disease, and chronic obstructive pulmonary disease. A two-tailed *p* < 0.05 was considered statistically significant. In models of multiple comparisons, the *p*-value was adjusted using the Bonferroni correction. All data were processed and analyzed using SAS software version 9.4 (SAS Institute Inc., Cary, NC, USA).

## 3. Results

The patient selection process is shown in Figure 1, and the characteristics of patients are shown in Table 1, stratified by classes. Out of 3932 non-operated AD patients, 10.8% (424 patients) were in group 0, 17.2% (676 patients) were in group 1, 26.3% (1035 patients) were in group 2, 28.0% (1100 patients) were in group 3, and 17.7% (697 patients) were in group 4. The mean age of all our study participants was 66.81 ± 14.82 years, and 71.29% (2803 patients) of them were men. The percentages of inpatient comedication of antiplatelets, anticoagulants, antidiabetic agents, and statins were 22.05%, 4.02%, 15.06%, and 14.47%, respectively. The most common comorbidities were hypertension (85.05%), hyperlipidemia (21.90%), and coronary artery disease (25.25%). Patients’ characteristics differed significantly between groups, showing that the group using more classes of antihypertensive drugs seemed to have a lower mean age, a higher proportion of male participants, and a higher proportion of hypertensive people. In most cases, AD locations were 68.41% in type B (UAD/AAD/TAAD) then 31.59% in type A (TAD).

Table 2 shows the prescription patterns in patients with non-operated AD during their 90-day post-discharge outpatient visit periods. Within the 10 categories of antihypertensive drugs, the three most prescribed types of antihypertensive drugs were CCBs (65.79%), β-blockers (62.46%), and ARBs (52.42%). In addition to the overall utilization of antihypertensive drugs, Table 2 also shows the combination of prescription patterns on the basis of classifying antihypertensive drugs into four main categories, including β-blockers, CCBs, RAS agents (ACEIs, ARBs, and renin-inhibitors), and other antihypertensive drugs. Within group 1, the most prescribed drugs were β-blockers (6.21%) and CCBs (5.57%). Within groups 2 and 3, the most prescribed drug combinations were β-blockers + CCBs (9.08%) and β-blockers + CCBs + RAS agents (14.95%), respectively.

Table 3 shows the univariate and multivariable Cox proportional hazard model analyses of different prescription patterns in non-operated AD patients within groups 1, 2, and 3. The event rate of different outcome are shown in Appendix A. Within group 1, compared to the control group, only RAS agents (aHR, 0.58; 95% CI, 0.39–0.84; *p* = 0.005) proved to have a significantly lower risk of occurrence of the composite outcome after adjusting for covariates selected by stepwise analyses and important risk factors for AD. CCBs (aHR, 0.72; 95% CI, 0.53–0.99; *p* = 0.043) also had a *p*-value of less than 0.05; however, after the Bonferroni correction for multiple comparisons, the difference was no longer statistically significant. Within group 2, the risk of the composite outcome was lower among patients using β-blockers + CCBs (aHR, 0.60; 95% CI, 0.42–0.85; *p* = 0.004) or CCBs + RAS agents (aHR, 0.60; 95% CI, 0.41–0.86; *p* = 0.006) than in the control group. Meanwhile within group 3, compared to the control group, there was no specific prescription pattern associated with a significantly lower risk of the outcome in the multivariable model.

## 4. Discussion

To the best of our knowledge, this is the first population-based study to investigate the effectiveness of antihypertensive drugs considering different combinations of prescribed antihypertensive drugs in non-operated AD patients. The characteristics of our study participants are similar to those of participants in a previous epidemiological study including non-operated AD patients (mean age, 67.6; male: female, 2.5:1) [10]. However, the mean age of our study participants was slightly lower. The possible reason for this observation is the increase in the prevalence of hypertension in the young and middle-aged population that has occurred over the past decade in our country [11,12].

In this study, the most prescribed antihypertensive agents were CCBs, followed by β-blockers and ARBs. Despite current guidelines suggesting that β-blockers are the mainstay of medical treatment for AD [2,5,6,7,8], β-blockers and CCBs were still most prescribed for blood pressure control in AD patients in Taiwan. This finding was also reported in previous studies investigating the prescription patterns in AD patients in Taiwan [10,13]. In addition, a review conducted in the past indicated that CCBs were the most prescribed antihypertensive agents in the management of hypertension in Taiwan, followed by ARBs [14].

For non-operated AD patients, the outcomes of exposure to RAS agents, β-blockers, and CCBs were improved, which are similar to the findings of this study. β-blockers have been shown to be associated with fewer AD-related events and lower growth rates of aortic aneurysms in single-center observational studies [15,16,17]. A previous study revealed that β-blockers were effective in reducing the rate of aortic-root dilation in patients with Marfan syndrome [18]. However, a study conducted in Japan evaluated the effectiveness of β-blockers, CCBs, and ACEIs using the database of the International Registry of Acute AD showed that the use of β-blockers was associated with a significantly lower risk of all-cause mortality; however, this was only in Type A and not in Type B [19]. Otherwise, there are some pieces of evidence suggesting the benefits of ACEIs and ARBs for AD treatment in patients with Marfan syndrome [20,21,22,23]. Furthermore, a recent study conducted by Chen et al. indicated similar benefits in that β-blockers, ACEIs, or ARBs were all associated with improved outcomes in AD patients [24].

The mechanisms of action of CCBs in the treatment of AD that go beyond blood pressure control remain unclear. CCBs were reported to reduce abdominal aortic aneurysm (AAA) expansion in an experimental model by inhibiting the expression of Matrix metalloproteinases (MMPs) [25,26]. However, a population-based observational study indicated that CCB use was an independent risk factor for the presence of an AAA [27]. Suzuki et al. [19] reported that β-blockers were associated with better survival only in Type A AD patients, while CCBs were associated with better survival only in Type B AD patients. Additionally, a single-center, retrospective cohort study conducted by Sakakura et al. including Type B AD patients reported that out of five types of antihypertensive drugs, only patients using CCBs had a significantly lower risk of all-cause mortality compared to those not using CCBs [28].

Per our findings, group 1 patients who were prescribed drugs acting on the RAS had a significantly lower risk of the composite outcome compared to those in the control group. Our findings in group 1 are comparable to those of the Chen study mentioned above [24]. As patients using both β-blockers and ACEI/ARB were excluded from the Chen study, the study design made it easier to compare the effectiveness of β-blockers, ACEI/ARBs, and other drugs since patients were only prescribed one class of antihypertensive drugs. In our study, within group 2, the combinations of β-blockers + CCBs (aHR, 0.60; *p* = 0.004) and CCBs + RAS agents (aHR, 0.60; *p* = 0.006) were shown to be associated with improved outcomes compared to the control group. Considering the abovementioned evidence, we hypothesized that the better effectiveness of these two drug combinations may partially come from the benefits of CCBs, which were seen especially in type B AD patients reported in the Suzuki study and the Sakakura study [19,28]. Non-operated patients in this study should mainly comprise type B AD patients because most of them are not eligible for surgical treatment (they are only eligible if they have complicated type B AD).

However, our study had certain limitations. First, to evaluate the overall effectiveness of antihypertensive drugs at the class level, we did not analyze the effectiveness of individual drugs or their dosages. Thus, the current study was limited to class effects and qualitative effects. Second, data on participants’ smoking status (a risk factor for AD), CT final reports, and details of the medical charts were not available from the NHIRD, therefore we were not able to review initial symptoms to know whether patients were complicated or uncomplicated AD in our survival rates. Third, secondary causes of AD, such as degenerative, sporadic, or traumatic dissections, as well as blood pressure values could not be detected with the NHIRD. Fourth, since it was a retrospective cohort study, the reason for certain medical treatments patients received was unknown, and this could lead to the misclassification of exposures. Finally, there may have been the risk of residual confounding. Despite having all these limitations, our study still provides substantial information about the treatment strategy for non-operated AD patients in clinical practice.

## 5. Conclusions

This nationwide retrospective cohort study demonstrated that the most prescribed antihypertensive drugs were CCBs, followed by β-blockers and ARBs, a finding which differed from the guideline recommendations of β-blockers as first-choice drugs for AD. Drugs acting on the RAS, β-blockers, or CCBs are effective antihypertensive drugs that should be prescribed in different treatment strategies to AD patients. Furthermore, further investigations should be performed to find an effective treatment to enhance clinical outcomes in non-operated AD patients.

## Figures and Tables

**Figure 1 jcm-12-01962-f001:**
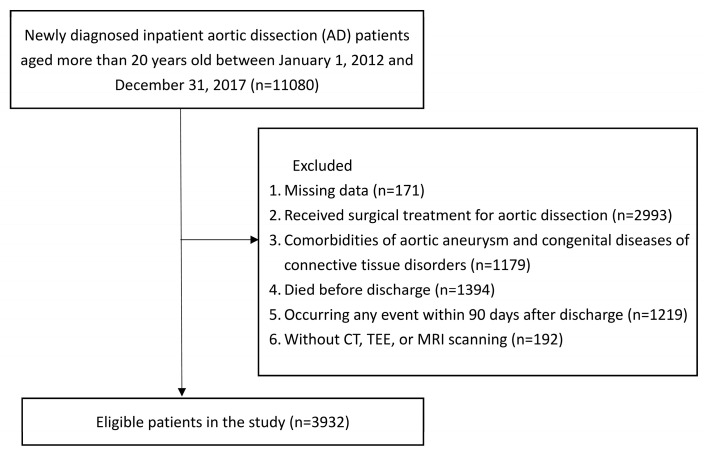
Flow chat of study population.

**Table 1 jcm-12-01962-t001:** Characteristics of Non-operated AD Patients, Stratified by Group.

Variables	Overall (n = 3932)	Group 0 (n = 424)	Group 1 (n = 676)	Group 2 (n = 1035)	Group 3 (n = 1100)	Group 4 (n = 697)
Age, mean, year (SD)	66.81 (14.82)	68.28 (16.55)	72.81 (13.87)	69.74 (13.48)	64.55 (14.12)	59.33 (13.79)
Sex, N (%)						
Male	2803 (71.29)	302 (71.23)	453 (67.01)	694 (67.05)	811 (73.73)	543 (77.91)
Female	1129 (28.71)	122 (28.77)	223 (32.99)	341 (32.95)	289 (26.27)	154 (22.09)
Geographic area, N (%)						
North	1737 (44.18)	176 (41.51)	297 (43.93)	462 (44.64)	480 (43.64)	322 (46.2)
Middle	695 (17.68)	85 (20.05)	133 (19.67)	171 (16.52)	180 (16.36)	126 (18.08)
South	1391 (35.38)	150 (35.38)	232 (34.32)	367 (35.46)	410 (37.27)	232 (33.29)
East	109 (2.77)	13 (3.07)	14 (2.07)	35 (3.38)	30 (2.73)	17 (2.44)
Urbanization, N (%)						
Urban	1866 (47.46)	203 (47.88)	304 (44.97)	495 (47.83)	514 (46.73)	350 (50.22)
Suburban	1636 (41.61)	177 (41.75)	289 (42.75)	428 (41.35)	466 (42.36)	276 (39.6)
Rural	430 (10.94)	44 (10.38)	83 (12.28)	112 (10.82)	120 (10.91)	71 (10.19)
Insurance premium, N (%)						
≤22,800 TWDs	2730 (69.43)	316 (74.53)	510 (75.44)	719 (69.47)	735 (66.82)	450 (64.56)
>22,800 TWDs	1202 (30.57)	108 (25.47)	166 (24.56)	316 (30.53)	365 (33.18)	247 (35.44)
Comedication, N (%)						
Antiplatelet	867 (22.05)	96 (22.64)	186 (27.51)	258 (24.93)	225 (20.45)	102 (14.63)
Anticoagulant	158 (4.02)	19 (4.48)	36 (5.33)	43 (4.15)	46 (4.18)	14 (2.01)
Antidiabetic agent	592 (15.06)	63 (14.86)	117 (17.31)	158 (15.27)	156 (14.18)	98 (14.06)
Statin	569 (14.47)	37 (8.73)	97 (14.35)	162 (15.65)	171 (15.55)	102 (14.63)
Comorbidity, N (%)						
Hypertension	3344 (85.05)	222 (52.36)	538 (79.59)	930 (89.86)	996 (90.55)	658 (94.4)
Hyperlipidemia	861 (21.90)	64 (15.09)	167 (24.7)	263 (25.41)	229 (20.82)	138 (19.8)
Diabetes mellitus	702 (17.85)	75 (17.69)	150 (22.19)	185 (17.87)	181 (16.45)	111 (15.93)
Heart failure	407 (10.35)	39 (9.2)	96 (14.2)	107 (10.34)	113 (10.27)	52 (7.46)
Atrial fibrillation	253 (6.43)	17 (4.01)	55 (8.14)	65 (6.28)	85 (7.73)	31 (4.45)
Coronary artery disease	993 (25.25)	87 (20.52)	222 (32.84)	293 (28.31)	265 (24.09)	126 (18.08)
Cerebrovascular disease	670 (17.04)	94 (22.17)	163 (24.11)	191 (18.45)	155 (14.09)	67 (9.61)
Chronic kidney disease	530 (13.48)	48 (11.32)	122 (18.05)	154 (14.88)	126 (11.45)	80 (11.48)
Chronic obstructive pulmonary disease	456 (11.60)	71 (16.75)	111 (16.42)	128 (12.37)	107 (9.73)	39 (5.6)
Charlson comorbidity index score, mean (SD)	2.38 (1.68)	2.68 (1.76)	2.91 (1.83)	2.46 (1.68)	2.16 (1.58)	1.93 (1.42)
Location of AD, N (%)						
UAD	666 (16.94)	-	-	-	-	-
TAD	1242 (31.59)	-	-	-	-	-
AAD	537 (13.66)	-	-	-	-	-
TAAD	1487 (37.82)	-	-	-	-	-

Abbreviations: TWD, New Taiwan dollars; SD, standard deviation; UAD, unspecified site of the aortic dissection; TAD, Thoracic aortic dissection; AAD, Abdominal aortic dissection; TAAD, Thoracoabdominal aortic dissection.

**Table 2 jcm-12-01962-t002:** Prescription Pattern of Non-operated AD patients.

Variables	Overall (n = 3932)
Categories of antihypertensive drugs, N (%)	
β-blocker	2456 (62.46)
CCB	2587 (65.79)
ACEI	174 (4.43)
ARB	2061 (52.42)
Renin-inhibitor	9 (0.23)
Diuretic	1062 (27.01)
Vasodilator	193 (4.91)
Centrally α_2_-agonist	41 (1.04)
α-blocker	664 (16.89)
Reserpine, Rauwolfia serpentine, guanethidine	8 (0.20)
Prescription patterns, N (%), stratified by classes	
Group 0	424 (10.78)
Group 1	676 (17.19)
β-blocker	244 (6.21)
CCB	219 (5.57)
RAS	107 (2.72)
Others	106 (2.70)
Group 2	1035 (26.32)
β-blocker + CCB	357 (9.08)
β-blocker + RAS	180 (4.58)
β-blocker + Others	88 (2.24)
CCB + RAS	246 (6.26)
CCB + Others	91 (2.31)
RAS + Others	73 (1.86)
Group 3	1100 (27.98)
β-blocker + CCB + RAS	588 (14.95)
β-blocker + CCB + Others	179 (4.55)
β-blocker + RAS + Others	123 (3.13)
CCB + RAS + Others	210 (5.34)
Group 4	697 (17.73)
β-blocker + CCB + RAS + Others	697 (17.73)

Abbreviations: CCB, calcium channel blocker; RAS, renin-angiotensin system. Others including diuretics, vasodilator (hydralazine, minoxidil, nitroprusside), centrally α2-agonists, α-blockers, reserpine, Rauwolfia serpentine, guanethidine.

**Table 3 jcm-12-01962-t003:** Univariable and Multivariable Cox Proportional Hazard Model Analysis for Composite Outcome.

	N	Events	PY	Rate (%) ^a^	Crude HR (95% CI)	*p* Value	Adjusted HR ^b^ (95% CI)	*p* Value
Group 1								
β-blocker	244	100	888	11.26	0.48 (0.35–0.66)	<0.001 *	0.74 (0.53–1.03)	0.074
CCB	219	110	741	14.84	0.64 (0.47–0.86)	0.004 *	0.72 (0.53–0.99)	0.043
RAS	107	47	402	11.70	0.51 (0.35–0.73)	<0.001 *	0.58 (0.39–0.84)	0.005 *
Others	106	68	290	23.46	1 (reference)		1 (reference)	
Group 2								
β-blocker + CCB	357	125	1452	8.61	0.42 (0.30–0.58)	<0.001 *	0.60 (0.42–0.85)	0.004 *
β-blocker + Others	88	43	310	13.87	0.67 (0.44–1.01)	0.057	0.69 (0.46–1.05)	0.086
β-blocker + RAS	180	69	672	10.27	0.49 (0.34–0.72)	<0.001 *	0.70 (0.48–1.02)	0.066
CCB + Others	91	59	296	19.92	0.96 (0.65–1.40)	0.814	1.11 (0.75–1.64)	0.617
CCB + RAS	246	92	949	9.69	0.47 (0.33–0.66)	<0.001 *	0.60 (0.41–0.86)	0.006*
RAS + Others	73	46	219	21.02	1 (reference)		1 (reference)	
Group 3								
β-blocker + CCB + Others	179	74	728	10.16	0.93 (0.68–1.26)	0.634	1.07 (0.77–1.47)	0.701
β-blocker + CCB + RAS	588	182	2343	7.77	0.71 (0.55–0.91)	0.008 *	1.01 (0.77–1.32)	0.942
β-blocker + RAS + Others	123	52	446	11.66	1.06 (0.76–1.5)	0.729	1.23 (0.86–1.74)	0.258
CCB + RAS + Others	210	89	811	10.97	1 (reference)		1 (reference)	

^a^ Rate was calculated as events divided by person-years; ^b^ Adjusting covariates selected by stepwise multiple regression analyses and important risk factors associated with aortic dissection, including age, gender, comorbidities of hypertension, hyperlipidemia, diabetes mellitus, heart failure, coronary artery disease, cerebrovascular disease, chronic kidney disease, chronic obstructive pulmonary disease. * *p* < 0.016 in group 1 and group 3; *p* < 0.01 in group 2 (adjust *p* value with Bonferroni correction for multiple comparison). Abbreviations: CCB, calcium channel blocker; HR, hazard ratio; PY, person-year; RAS, renin-angiotensin system; Others including diuretics, hydralazine, minoxidil, nitroprusside, centrally α2-agonists, α-blockers, reserpine, Rauwolfia serpentine, guanethidine.

## Data Availability

Corresponding author (Prof. Chung-Yu Chen) had full access to all the data in the study and takes responsibility for the integrity of the data and the accuracy of the data analysis. Data are available from the National Health Insurance Research Database (NHIRD) published by the Bureau of National Health Insurance (BNHI) of the Ministry of Health and Welfare. Owing to the legal restrictions imposed by the Government of Taiwan related to the Personal Information Protection Act, the database cannot be made publicly available. The conclusions presented in this study are those of the authors and do not necessarily reflect the views of the BNHI, the Ministry of Health and Welfare.

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
