# Peer review of "Evaluating Prescription Pattern and Effectiveness of Antihypertensive Drugs in Non-Operated Aortic Dissection Patients"

_jcm, 2023, doi:10.3390/jcm12051962_

Round 1
Reviewer 1 Report
This is a well written manuscript. It is clear, concise and easy to follow. The background and reasons for the study are well explained.
There are however some major issues in my opinion.
Abstract
Medical therapy in AD is a grade I recommendation (Riambau et al, EJVES 2017°
It is unclear, what is the control group ?
What is RAS?
Please divide the abstract section in subsection: introduction, materials and methods, results and conclusion
The conclusion is unclear and not very helpful
Introduction
L37: replace “associated”
L38: it is true for type B AD but no for type A AD, please clarify.
L51: please specify if it is type B AD or residual AD after type A repair or both? Complicated or uncomplicated AD?
Materials and methods
L66: please specify if it is type B AD or residual AD after type A repair or both?
L69: what kind of treatment? endoprosthesis? Surgical repair? …
L80: group 0 is the control group? if it is, please clarify.
L90: please remove this sentence, it is clear.
Did you have initial symptoms? it is important to know if it was complicated or uncomplicated AD ?
Please detail others antihypertensive drugs.
Results
There is a significant bias regarding the absence of anatomical analysis (aortic diameter, primary entry tear diameter, …). Indeed it is well known that an initial aortic diameter greater than 40 mm, a patent false lumen, a primary entry tear >10 mm , a false lumen partial thrombosis … were associated with risk of aneurysmal evolution, reintervention and death. These patients should have been removed from the analysis to study only the impact of medical treatment on the aortic dissection.
Please provide results for type AD and residual AD after type A repair separately.
Table 3: others: pleas detail
Discussion
It is well written, easy to read.
Conclusion
The conclusion is in accordance with the results but no very helpful.
Author Response
Response to Reviewer 1 Comments
Abstract
Point 1:
Medical therapy in AD is a grade I recommendation (Riambau et al, EJVES 2017).
It is unclear, what is the control group?
What is RAS?
Please divide the abstract section in subsection: introduction, materials and methods, results and conclusion
The conclusion is unclear and not very helpful.
Response 1: Thanks the reviewer’s suggestion. Due to the meaning of term and structure is easily to confusion for readers, we had modified all abstract and decreasing confusion, and make a clear and helpful conclusion. We also agree the reference Riambau et al, EJVES 2017, we had add the reference on revised manuscript.
Introduction
Point 2: L37: replace “associated”
Response 2: Thanks the reviewer’s suggestion. According reviewer suggestion, we had replaced associated.
Point 3: L38: it is true for type B AD but no for type A AD, please clarify.
Response 3: Thanks the reviewer’s suggestion. According reviewer suggestion, we had clarified the sentence.
Point 4: L51: please specify if it is type B AD or residual AD after type A repair or both?
Complicated or uncomplicated AD?
Response 4: Thanks the reviewer’s suggestion, it is very excellent suggestion for our study aim. According reviewer suggestion, we reword the aim and describe more clearly our study population in our study. In fact, our population had excluded type A endovascular aortic dissection repair, so there were only no operation and repair AD (including type A and Type B). Furthermore, there were limited information about CT final report or detail of medical chart in NHIRD database, we cannot have initial symptoms to know patients were complicated or uncomplicated AD in our survival. However, due to our study aim focused on over 90 days outcome for no-operation population, which may decreasing some bias when unknown complicated or uncomplicated AD.
Materials and methods
Point 5: L66: please specify if it is type B AD or residual AD after type A repair or both?
Response 5: Thanks the reviewer’s suggestion, we had described more clearly our study population in our study. In fact, our population had excluded endovascular aortic repair, so there were only no operation and repair AD (including type A and Type B).
Point 6: L69: what kind of treatment? endoprosthesis? Surgical repair? ....
Response 6: Thanks the reviewer’s suggestion. Due to the description was easily confusion for reader, we had modified the sentence. Furthermore, our study did not accept any invasive operation or repair in study population.
Point 7: L80: group 0 is the control group? if it is, please clarify.
Response 7: Thanks the reviewer’s suggestion. Due to the description was easily confusion for reader, we had modified the sentence. Furthermore, group only used for stratified population due to the risk bias occurring in different group.
Point 8: L90: please remove this sentence, it is clear.
Response 8: Thanks the reviewer’s suggestion. Due to the description was easily confusion for reader, we had removed the sentence.
Point 9: Did you have initial symptoms? it is important to know if it was complicated or uncomplicated AD ?
Response 9: Thanks the reviewer’s suggestion. Due to there were limited information about CT final report or detail of medical chart in NHIRD database, we cannot have initial symptoms to know patients were complicated or uncomplicated AD in our survival.
Point 10: Please detail others antihypertensive drugs.
Response 10: Thanks the reviewer’s suggestion. We had reworded on revised manuscript.
Results
Point 11: There is a significant bias regarding the absence of anatomical analysis (aortic diameter, primary entry tear diameter, ...). Indeed it is well known that an initial aortic diameter greater than 40 mm, a patent false lumen, a primary entry tear >10 mm , a false lumen partial thrombosis ... were associated with risk of aneurysmal evolution, reintervention and death. These patients should have been removed from the analysis to study only the impact of medical treatment on the aortic dissection.
Response 11: Thanks the reviewer’s suggestion. We are very agreed reviewer’s suggestion for excluding high risk of AD patients in our analysis. However, it is limited information about CT final report or anatomical analysis (aortic diameter, primary entry tear diameter, ...) in NHIRD. Moreover, due to we had excluded all operation and repair population, the final analysis population were only receive medical treatment of AD.
Point 12: Please provide results for type AD and residual AD after type A repair separately.
Response 12: Thanks the reviewer’s suggestion. We had provided overall population location of AD. From table 1, we can know that most of AD patients were type B (including AAD/TAAD/UAD) and only 30% were only medical treatment of TAA. Due to different group and combination had lower numbers for analysis, which causing bias for imbalanced combination, we did not analysis results of different AD.
Point 13: Table 3: others: pleas detail
Response 13: Thanks the reviewer’s suggestion. We had added described others in detail in table 3.
Discussion
Point 14: It is well written, easy to read.
Response 14: Thanks the reviewer.
Conclusion
Point 15: The conclusion is in accordance with the results but no very helpful.
Response 15: Thanks the reviewer’s suggestion. Due to the description was easily confusion for reader, we had reword conclusion in revised manuscript.
Reviewer 2 Report
Indeed an interesting study. Following are my suggestions, eventually searching to improve the manuscript.
Method.
- What type of AD are you dealing on? One can assume that your cohort is based on patients with type B AD (since type A Ad is always a surgical emergency), but you have not specified that, and I think it is worthwhile to do that.
- Were all your patients symptomatic for AD at its onset, or you also included incidental AD findings? I suppose all your AD patients were symptomatic, but you should specify that.
- You don’t specify the days of hospitalization, which is important at least to presume the stabilization of the AD patient from the acute (< 14 days) to the sub-acute phase (> 14 days).
Table 1.
- What is the Charlson comorbidity index? You should specify that.
- Is it really worthwhile to specify the geographic area of Taiwan where the AD patients come from?
Discussion.
Revision of the English language is necessary throughout this section: the meaning (and the message) of many paragraphs are not clear at all.
Conclusions.
Do not really reflect the results: you should be more specific.
Author Response
Response to Reviewer 2 Comments
Point 1: What type of AD are you dealing on? One can assume that your cohort is based on patients with type B AD (since type A Ad is always a surgical emergency), but you have not specified that, and I think it is worthwhile to do that.
Response 1: Thanks the reviewer’s suggestion. We had provided overall population location of AD. From table 1, we can know that most of AD patients were type B (including AAD/TAAD/UAD) and only 30% were only medical treatment of TAA. Due to different group and combination had lower numbers for analysis, which causing bias for imbalanced combination, we did not analysis results of different AD.
Point 2: You don’t specify the days of hospitalization, which is important at least to presume the stabilization of the AD patient from the acute (< 14 days) to the sub-acute phase (> 14 days).
Response 2: Thanks the reviewer’s suggestion. Due to our study population were no operation or endovascular aortic repair and there were stabilization when they leaved onset hospitalization (all of them over 14 days). Furthermore, our study evaluated effectiveness of antihypertensive drugs of 90 days prescription after hospitalization, we think days of hospitalization were not cause serve bias in our analysis.
Point 3: Were all your patients symptomatic for AD at its onset, or you also included incidental AD findings? I suppose all your, but you should specify that.
Response 3: Thanks the reviewer’s suggestion. It is limited information about CT final report or anatomical analysis (aortic diameter, primary entry tear diameter, …) in NHIRD. Moreover, we cannot have initial symptoms to know patients were complicated or uncomplicated AD in our survival. Furthermore, due to we only included inpatient at its onset, there is no incidental AD findings patients in our analysis.
Point 4: Table 1.
What is the Charlson comorbidity index? You should specify that.
Response 4: Thanks the reviewer’s suggestion. We had descripted in detail of CCI in revised manuscript.
Point 5: Is it really worthwhile to specify the geographic area of Taiwan where the AD patients come from?
Response 5: Thanks the reviewer’s suggestion. Due to different geographic area of Taiwan also indicated medical resources in Taiwan, most of NHIRD study in Taiwan will evaluate geographic area as covariate.
Discussion
Point 6: Revision of the English language is necessary throughout this section: the meaning (and the message) of many paragraphs are not clear at all.
Response 6: Thanks the reviewer’s suggestion. The entire manuscript has been corrected by a native English language expert.
Conclusion
Point 7: Do not really reflect the results: you should be more specific.
Response 7: Thanks the reviewer’s suggestion. Due to the description was easily confusion for reader, we had reword conclusion in revised manuscript.
Round 2
Reviewer 2 Report
Authors' explanations / revision of the manuscript are sufficient to accept the manuscript.
Author Response
Very thanks reviewer suggestion